**www.cambridge.org/ext**

Background extinction; extinction risk; functional extinction; mass extinction; biodiversity

**Corresponding author:**
P. David Polly;
Email: pdpolly@indiana.edu

# Extinction and morphospace occupation: A critical review

P. David Polly

Earth & Atmospheric Sciences, Biology, and Anthropology, Indiana University, Bloomington, IN 47405, USA

## Abstract

Processes of extinction, especially selectivity, can be studied using the distribution of species in morphospace. Random extinction reduces the number of species but has little effect on the range of morphologies or ecological roles in a fauna or flora. In contrast, selective extinction culls species based on their functional relationship to the altered environment and, therefore, to their position within a morphospace. Analysis of the distribution of extinctions within morphospaces can thus help understand whether the drivers of the extinction are linked to functional traits. Current approaches include measuring changes in disparity, mean morphology, or evenness between pre- and post-extinction morphologies. Not all measurements are straightforward, however, because morphospaces may be non-metric or non-linear in ways that can mislead interpretation. Dimension-reduction techniques like principal component analysis – commonly used with highly multivariate geometric morphometric data sets – have properties that can make the center of morphospace falsely appear to be densely populated, can make selective extinctions appear randomly distributed, or can make a group of non-specialized morphologies appear to be extreme outliers. Applying fully multivariate metrics and statistical tests will prevent most misinterpretations, as will making explicit functional connections between morphology and the underlying extinction processes.

## Impact statement

Whether extinction is random or selective is important for understanding the history of biodiversity and for better predicting the outcomes of anthropogenic extinction. Analysis of patterns of extinctions in morphospace can aid in understanding form and function interact with extinction processes in a selective way. Morphospaces are mathematical spaces constructed from variables that represent the form of organisms. If carefully constructed, the distribution of species in a morphospace summarizes their functional properties and ecological roles. The morphospace pattern of species that succumb to extinction can provide clues about the factors that make extinction more likely. This paper reviews strategies for analyzing extinctions in morphospace, explains some of the most common ways in which misinterpretations can arise from the mathematical properties of morphospace, and makes suggestions on how to avoid misinterpretations.

## Introduction

Whether extinctions are random or selective remains an important question in ecology and evolutionary biology. The standing diversity of species at any time ($t$) and place ($m$) is a balance between the rate of extinction ($\mu$) and the rate of origination ($\lambda$) such that $m_t = ae^{(\mu-\lambda)t}$, where $a$ is the standing diversity at an earlier time $t = 0$ (Raup, 1985). Thus, the nature of extinction – constant or episodic, random or selective, ecologically intrinsic or driven by external processes – is key to understanding the processes that control biodiversity past and future (Yule, 1925; MacArthur and Wilson, 1963; Raup and Sepkoski Jr, 1984; Raup, 1994; McKinney, 1997; Droser et al., 2000; Ciampaglio et al., 2001; Lyons et al., 2004; Koch and Barnosky, 2006; Roy and Goldberg, 2007; Jablonski, 2008; Jackson, 2008; Lockwood, 2008; Gill et al., 2009; Pereira et al., 2010; Alroy, 2015).

Morphospaces order species by their morphological traits in ways can be used to assess randomness or functional patterns by which taxa succumb to extinction. For example, morphospaces can help distinguish stochastic extinction from interspecific competition as in the Red Queen hypothesis (Van Valen, 1973) from non-random extinction linked features associated with trophic level, body size, geographic range, dietary or locomotor specialization, phylogenetic relationship, or physiological tolerance (e.g., Buzas and Culver, 1984; McKinney, 1997; Jablonski, 2008; Leighton and Schneider, 2008; Lockwood, 2008; Fritz and Purvis, 2010; Payne et al., 2016). Indeed, analyses of morphospaces themselves can reveal evolutionary constraints, many-to-one

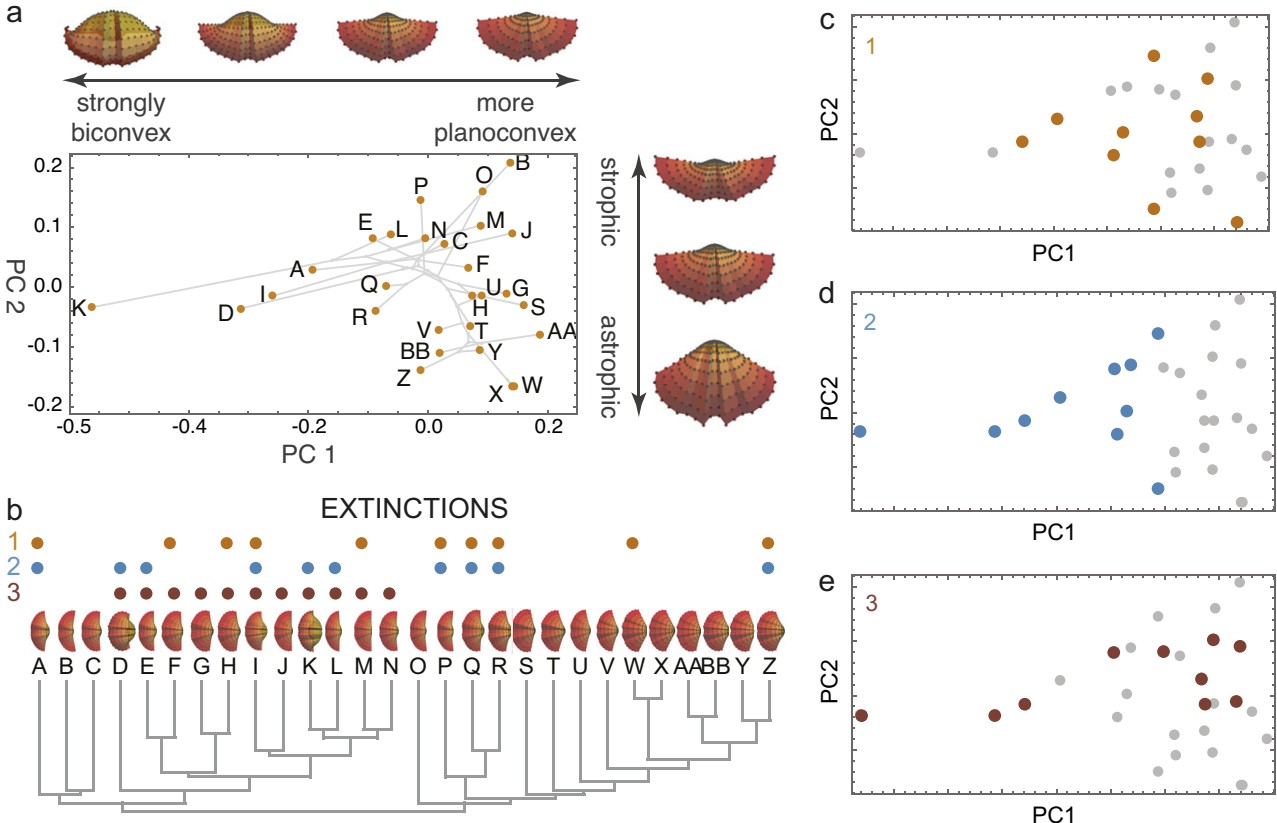

**Figure 1.** A simulated brachiopod morphospace (A) of valves evolved on a random phylogeny (B). The morphospace arranges the valves by convexity and hinge angle (strophic and astrophic). Three examples of extinction are illustrated: non-selective extinction (C), selective for strongly biconvex morphologies (D), and a selective by sub-clade (E). Colored dots in B–E show extinct species. Simulation follows procedures described by Polly and Motz (2017).

functional mappings, and patterns of convergence that may themselves feed into extinction processes (e.g., Raup, 1966; McGhee, 1999; Mitteroecker and Hetteger, 2009; Hallgrímmson et al., 2012; Chartier et al., 2014; Gerber, 2014, 2017; Polly, 2008, 2017). The increasing ease of obtaining morphometric data has allowed the role of morphological specialization in extinction to be more widely studied (e.g., Johnson et al., 1995; Hopkins, 2013; Wilson, 2013; Grossnickle and Newham, 2016; Halliday and Goswami, 2016; Hopkins and Gerber, 2017; Sibert et al., 2018; Polly, 2020; Bazzi et al., 2021; Ali et al., 2023).

Here I review the concept of morphospaces, ways of measuring patterns of extinction within morphospaces, pitfalls for interpreting patterns in high-dimensional morphospaces like those derived from geometric morphometrics, and remedies to avoid those pitfalls.

## Morphospaces and the study of extinction

A morphospace is any mathematical space defined by morphological variables. The simplest morphospaces are univariate, but they can have any number of dimensions defined directly by variables like length, width, and height, by transformed variables like principal components axes, or by axes derived from pairwise distances as in principal coordinates spaces (Thompson, 1917; Blackith and Reyment, 1971; Mardia et al., 1979; Mitteroecker and Hetteger, 2009; Chartier et al., 2014). Geometric morphometric morphospaces can have dozens or even thousands of dimensions.

Morphospaces can be derived theoretically from principles of embryonic development, geometry, or functional properties, or

they can be constructed empirically from a measured sample (McGhee, 1999). Raup's logarithmic shell coiling equations are a classic example of theoretical morphospace that represents mantle-based ontogenetic shell accretion using four parameters (aperture shape, whorl expansion, aperture translation, and the distance of the aperture from the coiling axis) to define a space of all possible shell shapes (Raup and Michelson, 1965; Raup, 1966). Most morphospaces, however, are derived from empirical data centered on the sample mean with unspecified limits of biologically plausible variation within their mathematically infinite bounds. Geometric morphometric morphospaces are empirical, as are multivariate spaces based on linear measurements or Fourier coefficients (e.g., Sokal, 1961; Rohlf, 1986, 1993). A simulated example of an empirical morphospace of brachiopods is shown in Figure 1. Rarely morphospace axes are based on categorical variables, such as Stebbins' (1951) floral space or Thomas and Reif's (1993) skeleton space. As discussed below, the mathematical properties of these morphospaces are varied – not all have orthogonal axes, not all are Euclidean, and not all are linear transformations of one another even when they are constructed for the same objects. Perceived patterns of extinction can therefore depend in part on the choice of variables and ordination.

Regardless, morphospaces order – or ordinate – species such that their spatial positions indicate similarity and differences that can be used to detect patterns of randomness or selectivity. Generalized statistical models of morphospace occupation exist that balance trait evolution, speciation, and extinction (Slatkin, 1981; Gavrilets, 1999; Pie and Weitz, 2005), as do studies of

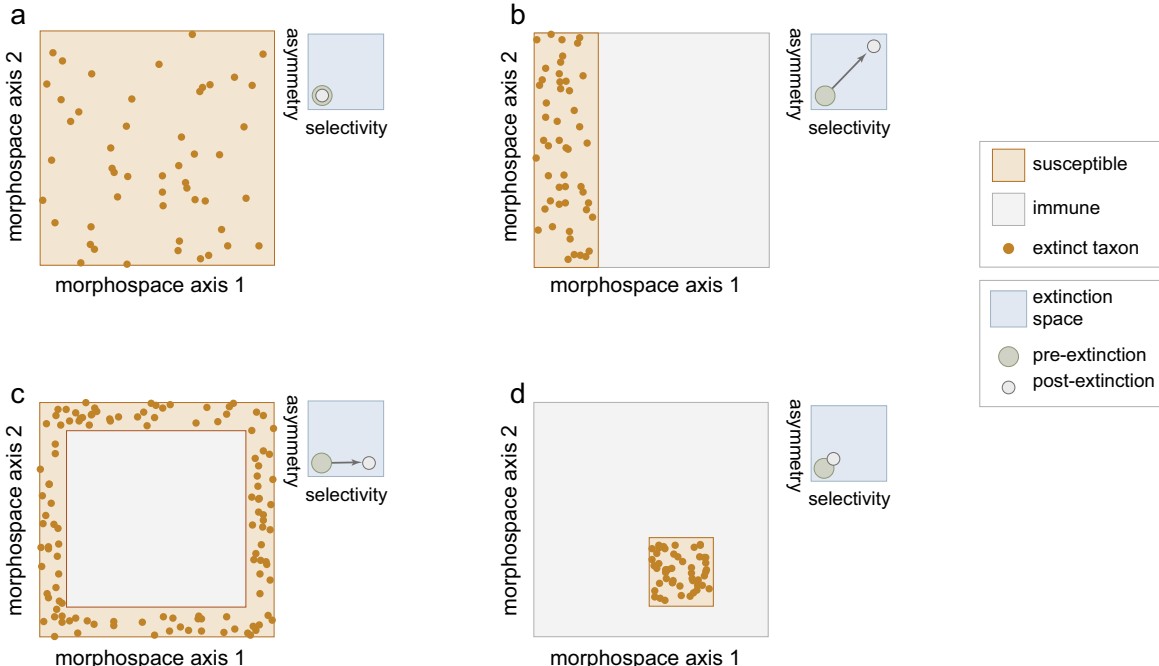

**Figure 2.** Four scenarios of how extinction affects morphospace distributions. Random extinctions (A) are spread stochastically across the morphospace leaving disparity high and mean morphology unchanged, corresponding to low selectivity and asymmetry in extinction space. Selective extinction at the negative end of one morphological axis (B) lowers disparity and shifts mean morphology, raising selectivity and asymmetry in extinction space. Selective extinction of all extreme morphologies (C) drops disparity drops but leaves mean morphology unchanged, corresponding to high selectivity but low asymmetry. Selective extinction of a non-peripheral subset of species (D) produces minor changes in disparity and mean morphology, with little or no change in extinction space.

statistical issues associated with measuring morphospace occupation (Ciampaglio et al., 2001).

## Extinction and morphospace distributions

By removing a subset of species, extinction transforms the distribution of taxa within morphospace, leaving a signature of the extinction process (Korn et al., 2013; Ali et al., 2023). Random (non-selective extinction) will reduce the total number of taxa (Figures 1C, 2A) but should have a little statistical effect on the moments of their distribution within the space. Selective extinction, however, may change the moments of the distribution, especially the mean and variance (variance in morphospace is one measure of morphological disparity (Foote, 1997)). If only certain kinds of specialized morphological outliers are more vulnerable (Figure 1D), both the disparity and the mean will be affected (Figure 2B), but if all morphologically distinctive species are likely to be culled, then the disparity will decrease but the mean will be unchanged (Figure 2C).

Non-selective extinctions in morphospace are expected not only under a truly random 'field of bullets' scenario (Raup, 1984), but also when the selective filter is unrelated to the variables that define the morphospace (Figure 1E), when the sample does not fully represent the range of selectivity of the extinction process (e.g., extinction differentially affects high trophic levels and only carnivores are included in the study), or under the Red Queen model in which all species are continually competing for limited resources and eventually lose (Van Valen, 1973). Selective extinction can occur when highly derived and ecologically specialized morphologies at the peripheries of morphospace are susceptible, when one part of the morphospace represents adaptations to an environment

that is hit by the extinction process, or similar scenarios. Contraction of niche space is an example cause of selective extinctions that would reduce morphological disparity (Valentine, 1995; Bush and Pruss, 2013). Geographic range size and niche breadth have been shown to be factors in selective extinction processes, but their connection to morphological traits (and thus morphospace) is indirect and varies from clade to clade (Jablonski, 2008, 2017; Harnik et al., 2012; Huang et al., 2015; Saupe et al., 2015).

As an example, Cole and Hopkins (2021) using morphospaces derived from discrete character data sets found that the Late Ordovician mass extinction of diplobathrid crinoids was random with respect to morphology and ecology, and that post-extinction recovery in this clade re-filled previously occupied regions of morphospace rather than exploring previously unrealized morphologies. The non-selectivity of the mass extinction was notably different from background extinctions in the same clade through the Paleozoic which selectively removed species with specific filtering strategies and habitats, especially those that were highly specialized (Baumiller, 2003; Liow, 2004). In contrast, Wilson (2013) found using morphospaces derived from geometric morphometrics that the end-Cretaceous extinction was highly selective on the dietary specializations of mammals, preferentially removing larger-bodied taxa with carnivorous and specialized herbivorous diets suggesting that the extinctions were caused by depressed productivity in the aftermath of the asteroid.

Korn et al. (2013) used the expected changes in disparity and mean morphology to distinguish selectivity and asymmetry in extinction. They used standardized versions of the range of morphological disparity, its total variance, and the change in the position of the mean morphology (centroid) to define an "extinction space". To take into account the correlation between range

and variance metrics, they used principal components to reduce their space to a "selectivity" axis driven by change in disparity and an "asymmetry" axis driven by shifts in mean morphology (Figure 2A–C).

Ali et al. (2023) pointed out that some extinction processes might selectively remove species that lie clustered within the overall distribution of taxa leaving a "hole" in the morphospace (Figure 2D). Disparity and mean morphology will usually be affected by such extinctions, but they will produce smaller changes on the selectivity and asymmetry axes of extinction space than when extinction differentially affects the periphery of morphospace. Nevertheless, selective extinction of morphologies on the interior of morphospace distributions could be just as easily produced as selective extinction at the edges by reductions of ecospace or culling of taxa at certain trophic levels.

Ali et al. (2023) also argued that random extinction may reduce disparity when species are concentrated near the center of morphospace (e.g., if they are multivariate normal). They therefore argued that the density distribution of species in morphospace is also an important metric for assessing extinction selectivity. If measured as variance, disparity would drop in this scenario only when the risk of extinction is distributed uniformly across the morphospace (because more rare species at the periphery would be lost than in the dense central region), but not if each species had an equal probability of extinction (range-based disparity would decrease in either scenario). Note that expectations are contingent on the peculiarities of the mathematical properties of the morphospace, the complexity of the distribution of species within it, and sample size (e.g., Ciampaglio et al., 2001). Regardless, the density distribution of species within a multivariate morphospace is more complex than it might appear as discussed below. Clumpiness or evenness statistics can be used to determine whether extinctions are clustered within limited regions of morphospace, regardless of whether at the periphery or in the interior (e.g., Heip et al., 1998; Tuomisto, 2012).

## Potential pitfalls

Gerber (2017) warned that the mapping between morphospaces derived from different quantifications of the same morphologies can be complex and nonintuitive. The topological relationships between alternative morphospace projections – the apparent relationships between taxa, the proportionality of their spacing, and their apparent location with respect to the center and periphery of the morphospace – is partly due to their morphology and partly due to the mathematical properties of the space. Careful attention should therefore be paid to how the morphospace is constructed and what can unambiguously be inferred from it.

Mitteroecker and Hetteger (2009) reviewed the mathematical and geometric properties of morphospaces constructed from several types of data and methods. Some are composed of many kinds of variables – linear caliper measurements, angles, volumes, areas, counts of structures, or matrices of meristic character states – that do not share a common unit of measurement. These spaces are referred to as non-metric because standard concepts of "direction" and "distance" are ambiguous: the scaling of one axis might be measured in radians and another in centimeters making the proportional relationship of the axes and the units of multivariate distance undefinable. Measures of evenness are likely to be quite ambiguous in nonmetric spaces.

The theoretical morphospace defined by Raup's shell coiling parameters (Figure 3A; Raup, 1966) is an example of a non-metric space whorl expansion, translation, and distance parameters are measured in different units making it difficult to equate a change in one direction relative to another. The morphospace defined by geometric morphometrics of the same shells, in contrast, is a metric space where a change in all directions is measured in the same Procrustes shape units (Figure 3B; Gerber, 2017; Polly, 2017). In the non-metric space, the measured disparity is dependent on arbitrary scaling between the incommensurate axes, but in the metric space the measured disparity is more objective. Measuring the asymmetric component of an extinction in non-metric space is problematic because the magnitude of a shift in one direction cannot be compared to a shift in a different direction, but asymmetry in metric space is invariant to direction.

Some spaces are metric but non-Euclidean (i.e., curved, bounded, or non-parallel; Mitteroecker and Hetteger, 2009). Constraints or covariances between variables can reduce the morphospace's dimensionality causing it to be curved or otherwise non-linear analogous to the surface of a sphere. Geometric morphometric spaces, for example, are non-Euclidean Riemannian hyperspheres whose dimensionality is reduced because of the translation, rotation, and scaling steps of Procrustes superimposition (Kendall, 1984; Dryden and Mardia, 1998). Distances and symmetry can be ambiguous in these spaces. In geometric morphometric morphospaces, shapes are identical at all of the spaces' peripheral edges, which would cause an extreme asymmetrical extinction (Figure 2B) to look like a peripheral model (Figure 2C).

More than one morphospace can often be constructed for the same species, like the snail shells in Figure 2. In this example, the same shell shapes are described by non-metric Raup coiling parameters in Figure 3A and by metric semilandmarks in Figure 3B. Both morphospaces are valid representations, but the mapping between them is non-linear (because of the logarithmic component of Raup's equations). The same pattern of extinction has different mathematical properties in each space: a uniformly random pattern in the Raup space would be nonuniform (i.e., selective) in the shape space and vice-versa; likewise, an extinction that culled the periphery of the Raup space would also cull the periphery in the shape space, but as a result the mean snail shell would change in shape space but not in Raup space. This seeming paradox is not a problem in the strict sense – it simply means that whatever the root cause of the extinction might be, it "sees" coiling geometry differently than overall shell shape – but the details are important for drawing interpretations about the extinction process from the observed pattern.

Multivariate morphospaces are common in studies of extinction, and they are usually constructed using either principal components analysis (PCA) from data consisting of metric variables or principal coordinates analysis (PCO) when the data are meristic or categorical. PCAs constructed from covariance matrices – appropriate when all of the variables have the same units like in geometric morphometrics – are simple rigid rotations of the data that maintain the original spacing and distance between the objects. In principle, PCA space is identical to the original variable space except for the coordinate system, but some consequences of the transformation may seem counter-intuitive when only two or three dimensions are visible. For more background on multivariate ordinations, including PCA, readers are referred to longer explanations in the literature (Tatsuoka, 1988; Hammer and Harper, 2006; Legendre, 2012; Polly et al., 2013; Polly and Motz, 2017). Properties like peripherality, position of the mean shape, and the distribution of species may differ markedly when the original variable space and PCA morphospace are viewed in typical two-dimensional projections.

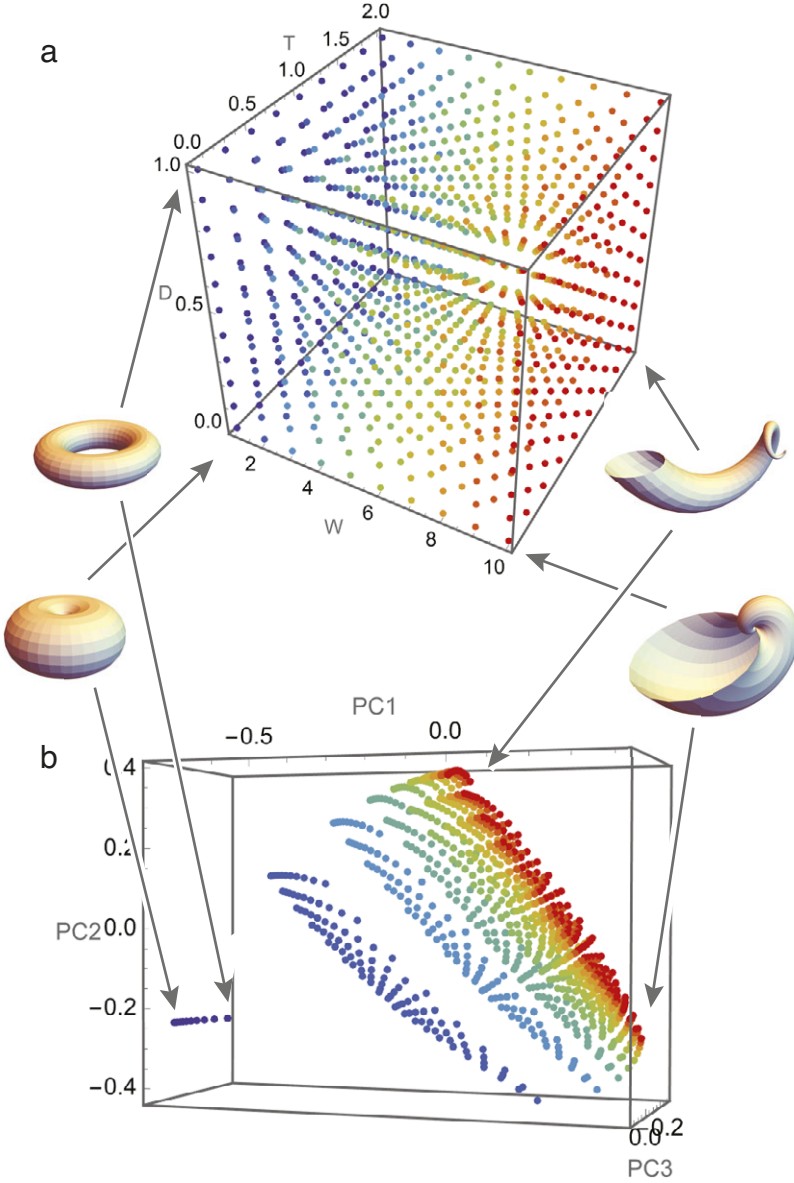

**Figure 3.** Two shell morphospaces: one expressed in Raup's shell coiling parameters (A) and the other expressed with geometric morphometric representations of the shell shapes (B). The color scheme of the points shows the position of the same shell in the two spaces (four peripheral points are illustrated with shells). W = whorl expansion rate; D = distance of whorl from coiling axis; T = rate of translation of whorl along coiling axis.

Figure 4 shows the first two dimensions (*X* and *Y*) of a five-variable space, each with a flat distribution that creates a rectilinear morphospace for the 10,000 random points shown here (Figure 4A, B). Because these variables have the same mean and variance, each accounts for about one-fifth (20%) of the total variance. When rotated to principal component (PC) space, the distribution projected onto the first two PCs counterintuitively appears to be circular (Figure 4C) with a greater density of points near the center of the morphospace (Figure 4D). Because the original variables of this example are uncorrelated, each PC axis also explains about one-fifth (20%) of the total variance. Despite being a rigid rotation, the relative distribution of points in PC 1 and 2 space appears to have no correspondence to their relative positions in the original *X* and *Y* space (Figure 4E).

The apparent differences between the original and ordinated PCA morphospaces are optical illusions stemming from the way

the multidimensional space is projected onto a plane. In the full dimensionality, the spacing between objects in the two spaces is identical (the PC space is a rigid rotation of the original variable space), but the axes of the first space are univariate, whereas the axes of the PC space are linear combinations of all five variables. By definition, the first PC axis is the axis of greatest variance in the original variable space, the second axis is orthogonal (at right angles) to the first and drawn through the next greatest axis of variance, and so on (Hotelling, 1933). The scattered points on the periphery of the PC space are those that have consistently high or low values on each of the original variables. The greatest Euclidean distance between two points in the original five-dimensional space is 22.67 (e.g., between a point lying at −10.0 on all five axes from one at 10.0 on all five) and it is the chance sampling of points that lie at distances close to this that define the principal component structure of this data set. Only a small number of points are

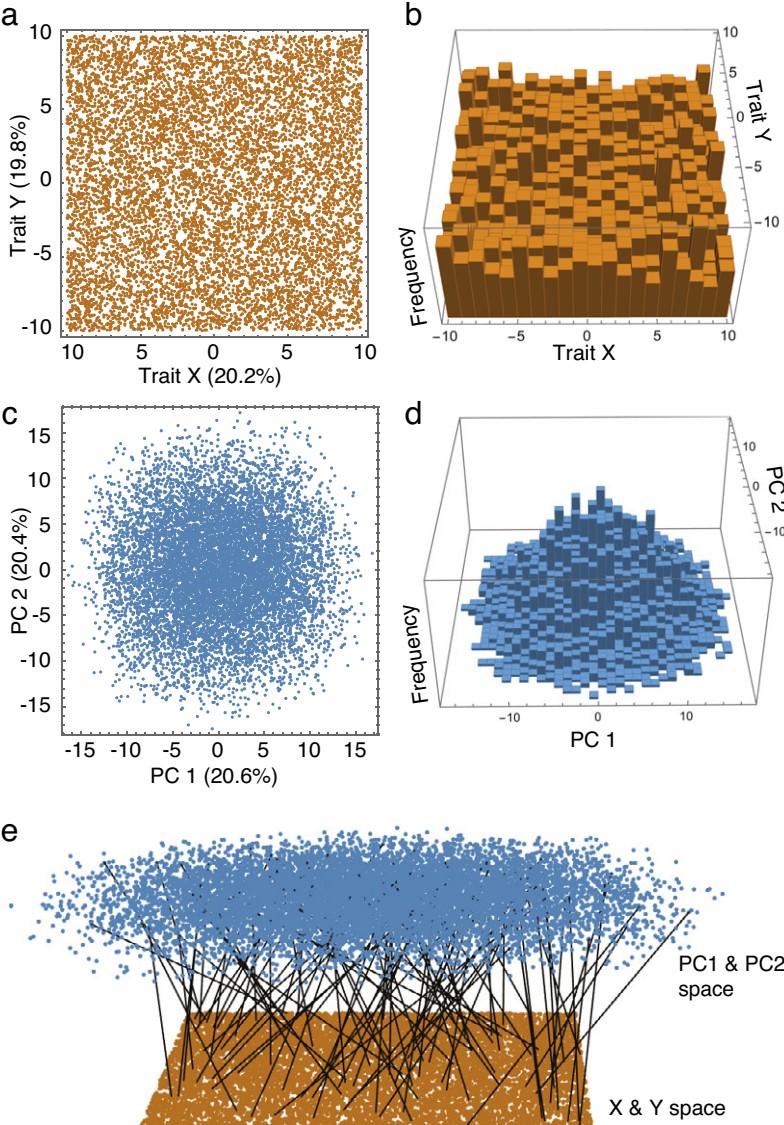

**Figure 4.** Comparison between the first two out of five dimensions of original variable space (A) where each variable has a uniform distribution (B) with a projection of the same data onto their first two principal components (PC1 & PC2) (C) where the density of points is greater near the center (D). A mapping of randomly selected points between the two spaces demonstrates that some that appear to be peripheral in the original variable space fall in the central region of PCA space and vice versa.

peripheral on all five variables, even though the frequency of points at the extreme (e.g., 10) is identical to the frequency of points precisely at the center (0) on any single variable. Most points that fall at an extreme on one variable fall between extremes on the other four. Consequently, the distribution of points *projected* onto the PC axes is denser at the center than at the periphery and points that appear to be on the periphery in a space defined by two of the original variables may lie in the center of the PC 1 and 2 space. The PC distribution appears circular simply by chance sampling; if 500,000 points had been used instead of 10,000 one would see that the edges of the PC distribution are actually straight, forming a polygon with four to ten sides depending on the orientation of the major axes of the sample, essentially the "shadow" of a five-dimensional hypercube cast in one direction or another.

The consequences of these transformational illusions are profound if an analysis of morphospace is conducted on only a subset of axes instead of the distribution in the fully multivariate space.

First, a finite sample measured in any subspace of a principal component morphospace (e.g., on the first two PC axes) is quite likely to have a higher density in the center regardless of whether the data actually have a flat or multinormal distribution. Tests for density or evenness should therefore be performed on *all* axes, not just the first few PCs. Second, measures of disparity and asymmetry also need to be assessed with all the axes, not just a couple. Figure 5 shows what three models of selective extinction with respect to the original variables look like projected onto the first two PCs and vice versa. What is actually a selective extinction of extremes on two of the five variables that reduce disparity appears to be randomly distributed in the space defined by the first two PCs with no change in overall disparity, but one that is actually peripheral on the first two PCs also appears to be peripheral (but fuzzier) in the original variable space (Figure 5A,B). Extinctions that are actually localized in the central part of either morphospace may appear to be asymmetrically peripheral in its counterpart space

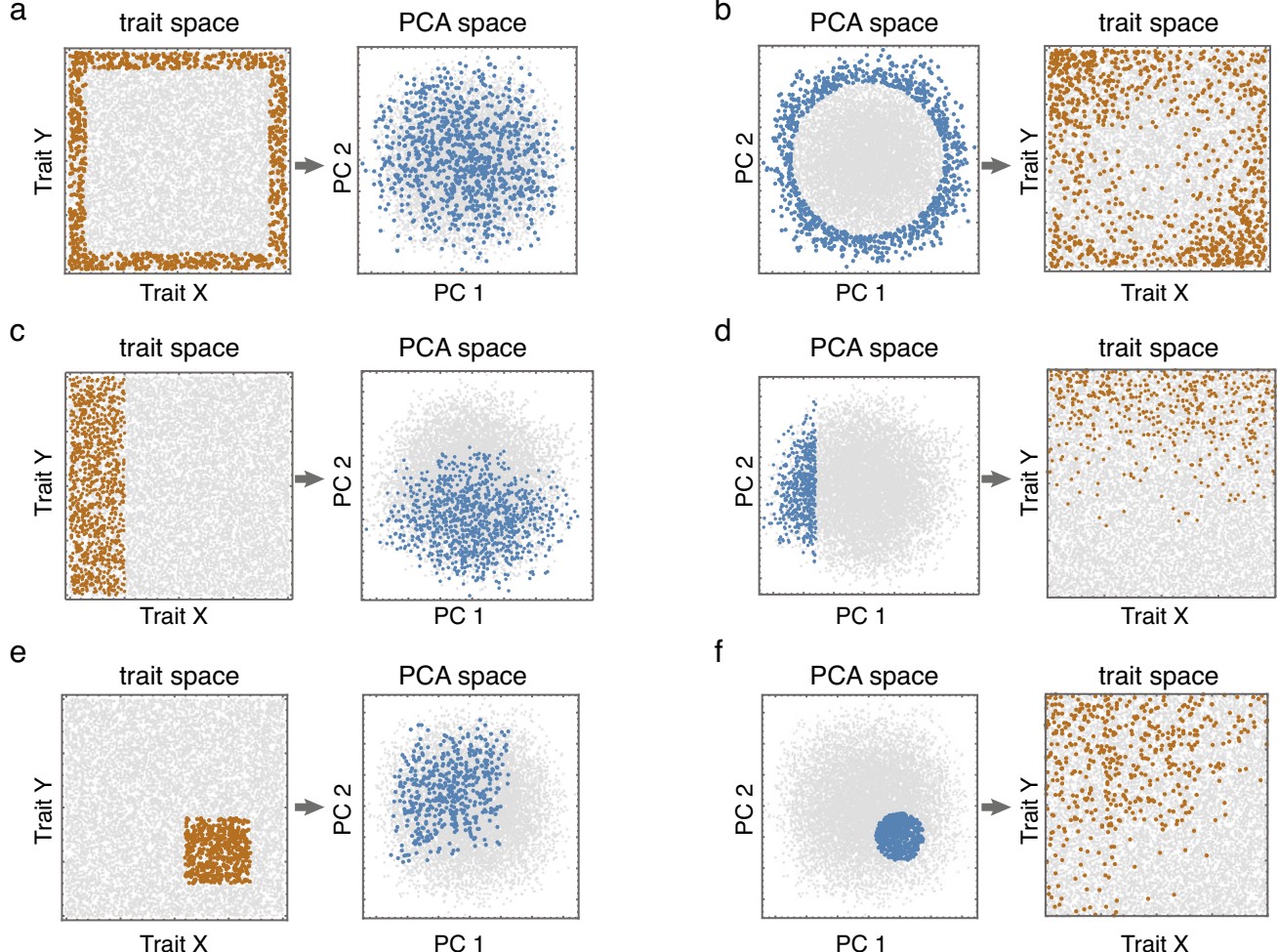

**Figure 5.** The apparent transformations of extinction selectivity between trait space and principal component (PCA) space when viewed in two dimensions. Extinction of peripheral morphologies in trait or PCA spaces can appear either random or fuzzily peripheral when transformed to the other space (A,B), asymmetrical extinctions may appear fuzzily asymmetrical (C,D), and extinctions of a small range of non-specialized morphologies may appear to affect a broad range of specialized morphologies (E,F).

(Figure 5E,F). In all cases, the seeming contradiction between the distribution of the same extinction in the two spaces is an illusion caused by viewing a multivariate distribution in just two dimensions.

## Conclusion

Because of the growing ease and speed with which morphological data can be collected (e.g., Boyer et al., 2015; Riley et al., 2015; Elder et al., 2018; Goswami et al., 2022), it is increasingly feasible to study processes of extinction using the lens of its selectivity within morphospace. From the distribution of extinctions in a morphospace it may be possible to infer the causal links between extinction processes, environments, and organismal function that will lead to a better understanding of what differentiates background extinctions from the escalating events that produce of mass extinctions.

Morphospaces can have mathematical ambiguities that may confound interpretations, however, including their metric properties, non-linear mappings between morphospaces represented by different sets of variables, and the distorting effects of dimension reduction techniques. Many of potential pitfalls can be easily circumvented. Several authors have made recommendations how to avoid the potentially misleading consequences of ignoring the full

dimensionality of morphospace (Bookstein, 2013, 2016; Polly et al., 2013; Goolsby, 2015; Uyeda et al., 2015; Polly and Motz, 2017; Adams and Collyer, 2018; Cardini et al., 2019). Disparity, asymmetry, and evenness statistics performed multivariately on all five dimensions of the space should produce identical results if calculated on the original variables or the PC scores; fully multivariate tests will therefore get around most problems.

Arguably, interpretations about extinction processes are most effectively framed in terms of biological or ecological significance of the specific variables that define the space rather than on the general pattern – some variables may be relevant to the selective extinction process, others may be correlated with the relevant variables, and yet other variables may be random (uncorrelated) with respect to those that are. Focusing on the functional roles of the morphology will also circumvent many of the mathematical ambiguities between different projections of shape space described above. For example, Hebdon et al. (2022) recently used performance spaces, in which multivariate shape is regressed onto independent measures of functional performance to estimate functional gradients in the shape space (Polly et al., 2016), to study the selectivity of the Triassic–Jurassic extinction relative to swimming performance and life strategies in ammonites. If carried out multivariately, such an approach would come to the same conclusion about whether the

extinction was selective relative to morphological function regardless of which projection of morphospace was used, or even whether a Raup space or geometric morphometric had been used.

**Open peer review.** To view the open peer review materials for this article, please visit http://doi.org/10.1017/ext.2023.16.

**Data availability statement.** No original data are presented in this paper. Simulated data were generated with code that is publicly distributed (Polly, 2022a, 2022b, 2023).

**Acknowledgements.** Thanks to John Alroy and Barry Brook for suggesting I write this paper, to Elizabeth Housworth for answers about mathematical spaces, to Alycia Stigall for advice about brachiopod morphology, and to two anonymous reviewers, Jonathan Payne, Wolfgang Kiessling, and Laetitia Black for editorial support.

**Author contribution.** P. David Polly is the sole author of this paper.

**Competing interest.** The author declares no competing interests exist.

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
