## [Reviewer Report]

The manuscript is both topical and informative. I found it an interesting read, but I do have a few recommendations for the author.

1. The manuscript builds on/ compliments Ciampaglio et al. 2001 paper "Detecting changes in morphospace occupation patterns in the fossil record: characterization and

analysis of measures of disparity" but is not mentioned in the paper.

2. The author mentions “fireballs from the sky” as a reference to Mass Extinctions (no doubt the End Cretaceous Event). This is a somewhat glib interpretation of what caused the End Cretaceous extinction event. Mass extinctions are events that are quite complex, with many factors involved. As such I would refrain from over-simplification.

3. I think the section on the correspondance of axes based on variables (both meristic and metric) to those of PCA and PCO spaces should be more thoroughly explained.

4. I think that an example using a real, but simple data set burrowed from the literature, or created by author, would help the reader better understand the the correlation between meassured characters and disparity.

---

## [Reviewer Report]

This is an insightful review about how the dynamics of taxa distribution in morphospace could reflect extinction selectivity (or the lack of it). The author laid out the various scenarios, discussed important analytical strategies as well as potential pitfalls. Constructive suggestions were also provided at places. Overall, I think this is a useful synthesis that would fit very well with the journal and is highly relevant to the development and major future directions in the field of macroevolution.

I only have some minor comments, listed below, using the line numbers (L) in the PDF.

L5-6 (Abstract): a typo or missing word in “Analysis of the patterns extinctions...”

L20 and later in L163-165: it might be worth pointing out that these arguments depend on the standing diversity and extinciton intensity. A normal distribution gets shaky when the number of taxa (or those that survived) is low and the change in variance (and probably range) after random extinction will be increasingly variable.

L51-64: the intro is very insightful but I’d also appreciate a bit highlight on why it is important to look at extinciton in relation to morphospace or morphology beyond body size and other ecological properties. I think this should even come before the technical ease and data availability, e.g. something along the lines of morphology allowing but also constraining functions and morphospace dynamics being interesting in themselves as important mechanisms of extinction. It is important to ecognize mismatches between morphological variation and functional diversity in macroevolution, which I think should be acknowledged (if not highlighted) in the intro and the conclusion.

L121-122: this scenario doesn’t sound symmetrical to me or did you mean this as an assymmetric case?

L124-125: there’s a jump in the focal level; here it’s about geographic range and niche breadth of the individual taxa, as represented by the points in the figure but the preceding point was about niche space of the whole clade, as represented by the whole space in your figure. these should probablly be clarified to avoid confusion.

L182: it might be helpful to explain right away what Euclidean versus non-Euclidean mean.

L210: “to” look like?

---

## [Editor Report]

The reviewers both found the manuscript to be well written and clearly illustrated. They have just a few minor comments that should improve the manuscript once addressed. I agree with this assessment. Illustrating results from a real data set could be quite informative for readers, if space allows.

Line 53: “non-random selective” seems redundant to me

Line 97: missing “of” between “models” and “morphospace”

---

## [Editor Report]

The author has done a thorough and responsible job of addressing all reviewer comments. The decisions about how to address conflicting recommendations or those that would send the manuscript beyond the length limit are all reasonable. I concur with the reviewers in recommending that the manuscript be accepted in its current form.